# The Influence of Sociodemographic Factors on Symptoms of Anxiety, the Level of Aggression and Alcohol Consumption in the Time of the COVID-19 Pandemic among Polish Respondents

**DOI:** 10.3390/ijerph19127081

**Published:** 2022-06-09

**Authors:** Szymon Florek, Magdalena Piegza, Paweł Dębski, Piotr Gorczyca, Robert Pudlo

**Affiliations:** Department of Psychiatry, Faculty of Medical Sciences in Zabrze, Medical University of Silesia, 42-612 Tarnowskie Góry, Poland; mpiegza@sum.edu.pl (M.P.); pdebski@sum.edu.pl (P.D.); pgorczyca@sum.edu.pl (P.G.); rpudlo@sum.edu.pl (R.P.)

**Keywords:** aggression, alcohol consumption, anxiety, COVID-19

## Abstract

The COVID-19 pandemic has had a significant impact on the mental health of most of the world’s population. The authors of this study decided to identify differences in the intensity of anxiety, aggression and alcohol consumption within the study population. The study was conducted via an Internet survey. It uses Polish adaptations of international scales to assess anxiety (GAD-7), aggression (Buss and Perry Aggression Scale) and alcohol consumption (AUDIT test). A total of 538 people were examined. The surveyed group of respondents does not reflect the structure of Polish society. The group of surveyed women scored higher than men in terms of anxiety, as well as anger and hostility. The examined men were characterized by higher results of alcohol consumption and physical aggression. People between 18 and 49 years of age consumed significantly more alcohol than people aged 50 and over. People aged 18–29 obtained statistically significantly higher scores in generalized aggression and hostility. The relationships between the studied parameters do not differ significantly from those presented in other studies. People under the age of 50 are the group most exposed to the harmful effects of alcohol. Women between the ages of 30 and 49 are most vulnerable to the negative mental health effects of the COVID-19 pandemic. There is a need for further research studies in which the studied group will obtain a greater degree of compliance with the structure of Polish society.

## 1. Introduction

The COVID-19 pandemic has significantly affected the mental health of societies in various regions of the world [1]. The researchers have observed an increase in the level of both domestic [2] and intimate [3] aggression. The Portuguese study on domestic violence in the COVID-19 era highlights the fact that most victims were unable to find help, indicating the need to make further investments in support systems that would provide good care for victims of violence [2]. There are many studies on the fear of COVID-19 infection. In one of them, a special scale was presented to measure the level of anxiety in relation to COVID-19 [4]. In another study conducted in China, researchers showed that 14% of the people they surveyed struggled with anxiety during the COVID-19 pandemic [5]. There are also studies on the amount of alcohol consumed during this period. Research conducted in Poland shows that 14% of the surveyed people increased the amount of alcohol consumed during the COVID-19 pandemic [6].

The studies presented above leave no doubt that the period of the COVID-19 pandemic affects mental health in many different societies. It is worth noting, however, that this impact can be understood in two ways: First, it can be understood as an immediate threat of contracting SARS-CoV-2 or your loved ones, which can be fatal. In this approach, COVID-19 has the greatest effect on increasing the level of anxiety [4,5]. Secondly, one should take into account that the impact of the lockdown introduced by the government, which significantly limited the civil rights of society [7,8], caused specific isolation of the individual and its legal justification was subject to many different deliberations [9]. This situation undoubtedly influenced the mental health of citizens [10]. Many studies have confirmed the impact of lockdown on the escalation of aggressive behavior—which is also widely discussed in the philosophical and legal environment [11], as well as on increasing anxiety [10] and alcohol consumption [12]. In addition, it is necessary to mention the mechanisms between individual variables and, thus, above all, the intensification of both aggression and alcohol consumption by fear, which also takes place during the COVID-19 pandemic [13].

Due to the fact that the survey concerns Polish respondents, it seems reasonable to analyze the behavior of Poles during the first wave of the COVID-19 pandemic, which is in the first half of 2020. Firstly, we need to examine Polish culture and its geographical and political conditions. Studies on the impact of the pandemic on the mental health of Poles indicate a deterioration in the quality of sleep, especially among young people [14], an increase in anxiety and depressive disorders and a sense of loneliness among women, especially for young women [15]. In the geographic-political context, a large international survey was conducted with 9543 respondents from 11 countries around the world. It showed that Poles have the lowest amount of trust (of all nationalities surveyed) toward health care and the lowest amount of trust relative to whether the government will take care of its citizens, and similar results were obtained from Bulgaria and Romania. Moreover, it was in these three countries that the governments decided to implement the most severe restrictions related to the first wave of the SARS-CoV-2 pandemic [16]. One such limitation has certainly been poor accessibility for patients to medical care. According to the report by the Polish Ombudsman for Patients, March 2020 was the month in which the highest number of telephone complaints were received from patients. Most of them generally concerned limited accessibility to health care services. By analyzing the complaints in more detail, it can be seen that most of them concerned primary health care, followed by outpatient specialist care and hospital treatment. Importantly, the three most frequent reasons for complaints about outpatient specialized care related to problems with registration, waiting times for services and access to teleprompters [17]. Admittedly, there are no data for individual specialties, but the authors are fully aware that outpatient specialist psychiatric care is also among the data analyzed. Of course, in this situation, special focus should be directed toward other forms of help related to mental health. As the conducted research shows, during the COVID-19 pandemic, the number of people using various religious practices increased [18,19]. Boguszewski et al., however, noted that this increase may be temporary, even delusional, as the limitations of the COVID-19 pandemic may help some people opt out of regular religious practice [19]. A completely opposite thesis was presented by Kowalczyk et al., suggesting that in response to anxiety and a sense of suffering, a “coronavirus generation” may arise, which they say would experience “spiritual renewal” [18]. Another study pointed to a certain risk associated with having strong faith. Namely, according to the authors, prayer can actually help in coping with the stress of the COVID-19 pandemic, but larger religious practices can in themselves be stressful and, therefore, dangerous to the human psyche. Moreover, the symptoms of depression are in no way modified by the discussed practices [20]. In contrast to the symptoms of depression, religion turned out to be significant in the context of the severity of anxiety. It turns out that it increased much more among Catholic women than among others [21]. Finishing the topic of religion and its influence on behavior in the context of the COVID-19 pandemic, it should be noted that subsequent studies showed a potential relationship between a high level of religiosity and greater fears (for health, finances and employment), worse mental wellbeing and a tendency to believe in controversial theories about the new coronavirus. It has also been proven that activities aimed at building trust in health protection and pharmaceutical companies should be directed, in particular, to people who frequently practice religion [22,23].

This study aims to identify differences in the group of respondents participating in the online survey in terms of the intensity of symptoms of anxiety, aggression and alcohol consumption depending on age, gender, education and place of residence. We will also try to check whether the pandemic affects the interrelationships between the factors studied. Based on the results obtained, it will be possible to distinguish social groups that are particularly exposed to the negative health consequences of the COVID-19 pandemic. It should be emphasized that this study is part of a larger project aimed at examining the relationship between anxiety, aggression, alcohol consumption and mental resilience during the time of the COVID-19 pandemic.

As mentioned above, one of the aims of this article is to analyze the level of alcohol consumption by respondents during the SARS-CoV-2 virus pandemic. Thus far, many different studies have been conducted on this issue in Polish society. One of them, the PURE Poland cohort study conducted in the Lower Silesia Voivodeship, allowed the determination of some socio-demographic determinants of alcohol consumption. Inhabitants of cities with higher education are most likely to drink low alcoholic beverages, whereas inhabitants of rural areas and people with lower education are more likely to choose high alcoholic beverages. Furthermore, the study found that men are significantly more likely to use alcohol than women, and they tend to drink heavily, which is consistent with studies conducted around the world, including in Europe [24,25,26]. The present study also analyzed the age of the study participants, thanks to which it was shown that young people under the age of 45 are much more likely to use alcohol [24]. The research conducted to date also points to the cultural and geographic context associated with alcohol consumption. The Polish drinking pattern is relatively similar to that in Eastern countries, such as Lithuania, Belarus or Russia. Although Poland and Lithuania consume less alcoholic beverages than the other two countries, research shows that alcohol is still a cause of many deaths [26,27].

The study was conducted in the period from 24 April 2020 to 8 May 2020. At that time, 4855 people fell ill in Poland, 322 of whom died. These data come from the official statistics of the Polish Ministry of Health available on the government website gov.pl (accessed on 24 April 2022) and the official profiles of the Ministry of Health on websites such as Twitter and Facebook.

## 2. Materials and Methods

The study was entirely based on an online survey disseminated via online channels such as Facebook and other social networks. Such a procedure was mainly due to the epidemiological restrictions that were in force in Poland at that time. The collection of data for the study was carried out in compliance with all rules of confidentiality. At the beginning of the survey, there was comprehensive instruction explaining the purpose of the study and its legitimacy. It also contained information on the possibility of withdrawing from participation in the study at any possible time. In order to complete the questionnaire, the respondent had to previously accept described rules. The Bioethics Committee at the Medical University of Silesia in Katowice concluded that the study did not require its consent. The inclusion criterion for the conducted study was 18 years of age. The survey was structured in such a way as to include all the exclusion criteria at the beginning, which were as follows:The presence of significant life changes (such as a change of job, marriage, divorce or the birth of a child) in the last 12 months before the study;Constant care of a psychiatrist.

All the above conditions were met jointly by 538 people, including 413 women (76.77%) and 125 men (23.23%). However, all subjects were asked to answer a question about concerns related to the current COVID-19 pandemic. There were 1093 respondents in total, including 866 women (79.23%) and 227 men (20.77%). The exact structure of the study group that met the inclusion criteria is presented in Table 1.

Respondents who met the inclusion criteria for the study answered the questions included in the AUDIT (*Alcohol Use Disorders Identification Test*), GAD-7 (*Generalized Anxiety Disorder-7*) and the Buss–Perry Aggression Scale.

### 2.1. Scales Used

The Polish adaptation of the AUDIT scale [28] consisting of 10 questions was used. It helps identify problems related to alcohol use; however, it is not a substitute for a clinical examination, and it is not possible to diagnose addiction solely on its basis. The questions are scored on a scale of 0–4. The final score is calculated by adding up all the points scored. The Cronbach’s alpha coefficient in our study was 0.79.

The GAD-7 scale, originally designed by R.L. Spitzer, J.B.W. Williams, K. Kroenke and colleagues [29], was used to determine the intensity of anxiety. The Polish adaptation of this scale contains a total of 7 questions. Respondents answering questions on this scale were asked to relate them to the last two weeks. The total score is the sum of all points. The Cronbach’s alpha coefficient in our study for the GAD-7 scale was 0.87.

In order to measure the level of aggression, the Polish adaptation of the A.H. Buss and M. Perry aggression scale was used [30,31]. The answer to each question was assigned a certain number of points on the Likert 5-point scale. The final score was obtained by summing up all points (two questions are scored oppositely). It presents the level of generalized aggression, which includes the subscales of anger, verbal aggression, physical aggression and hostility. The Cronbach’s alpha coefficient in our study is slightly higher than in the validation work—it amounted to 0.88 [31].

### 2.2. Statistical Analysis

Statistical analysis was carried out using Excel 2016 and Statistica version 13.3 (TIBCO, Hamburg, Germany). The Shapiro–Wilk test was used to assess the normality of the data distribution. To check the differences between the studied socio-demographic groups, the Mann–Whitney U and Kruskal–Wallis ANOVA tests were used. The chi-square test was used to check the differences between the groups in the results obtained after normalization. The internal consistency of the applied scales was estimated using Cronbach’s alpha coefficient.

## 3. Results

Comparative analyses carried out within the study group showed statistically significant differences between the groups of women and men, as well as older and younger people concerning the parameters of the mental state under study. On the other hand, the place of residence did not influence the obtained results.

The comparative analysis between the groups of women and men about individual parameters is presented in Table 2.

It can be concluded that women obtained higher scores than men in terms of perceived fear, as well as in two subscales of aggression—anger and hostility. The group of surveyed men was, in turn, characterized by higher results in terms of the level of alcohol consumption and physical aggression. There were no statistically significant differences between men and women in terms of verbal and generalized aggression. For a more accurate presentation of the results, the commonly accepted norms for the psychometric scales of the level of alcohol consumption and anxiety were used, as shown in Table 3.

Table 4 shows the differences in the assessed indicators depending on age.

The analysis shows that people aged 18–49 consumed significantly more alcohol than people aged over 50. Statistically significant differences were also noted in generalized aggression and one of its subscales—hostility. In both cases, the results were higher in the group of people aged 18–29. Table 5 summarizes the results of the scales for anxiety and alcohol consumption in relation to age. There are no differences between groups according to both parameters—anxiety and alcohol consumption.

Comparative analyses were also made for the breakdown of the study population according to education. The obtained results showed a statistically significantly higher level of generalized aggression and its three components—verbal aggression, anger and hostility—among people with secondary education (Table 6). It should be noted that the comparison was possible only between people with secondary education and higher education. Table 7 presents the results of the scales for anxiety and alcohol consumption in accordance with recognized standards that are broken down by education level.

All respondents (*n* = 1093) were asked to answer a multiple-choice question about pandemic concerns. The results from the analysis of this parameter are presented in Table 8.

## 4. Discussion

The analysis of the results showed that anxiety is stronger in women than in men. This finding is in line with other studies conducted outside the pandemic period. They indicate that women are more susceptible to anxiety-depressive disorders, which is part of the common thinking about female psychopathology; however, the reason for such a state of affairs has not been clearly defined, and the studies conducted so far indicate its multifactorial nature [32].

In our study, the group of respondents achieved statistically significantly higher scores in the subscales of anger and hostility with male respondents. These results differ slightly from the studies conducted so far [30,33]. Firstly, in the study by Buss and Perry, the group of women obtained slightly lower values in the subscale of hostility towards men, while in terms of anger, no significant differences between the sexes were found [30]. Nevertheless, the study conducted among Argentinian adolescents showed that anger was significantly higher in the group of women, while in terms of hostility, the authors did not find a statistically significant differences between the studied women and men [33]. These differences may be due to many reasons, including cultural differences, the way the study was conducted, the methods used, socio-demographic differences and finally the lack of a pandemic, which in our study was the reason for its initiation.

Statistically significant differences between men and women in terms of physical aggression measured with the Buss and Perry scale were confirmed by the two above-cited studies [30,33]. They are consistent with the results obtained in this study, indicating that the group of male respondents is characterized by significantly higher values in the subscale of physical aggression.

In the AUDIT test, statistically significantly higher results were obtained by the group of the examined men. This result—similarly to the previous one—is identical to that obtained from other studies before the pandemic period [24,25,26,34]. As the study from the time of the first wave of the COVID-19 pandemic shows, this pattern in general—as presented in this study—has not changed in principle. Nevertheless, the article in question shows an increase in wine consumption with a decrease in the consumption of spirits among both men and women. Moreover, women in this study were more likely to report increased levels of wine consumption compared to male respondents [35].

The analyses also showed that both hostility and generalized aggression are statistically higher in the group of the youngest respondents aged 18–29. This may be related to the need to vent the frustration resulting from the restrictions imposed due to the nationwide lockdown. However, this observation requires further investigation in the later stages of the pandemic.

Our study also shows that statistically more alcohol is consumed by people between 18 and 49 years of age compared to those over 50. However, when analyzing the data obtained after the normalization of the results, no changes in alcohol consumption were found. This state of affairs is reflected in the first raw data analysis, where the mean scores of both younger and older people ranged from 0 to 8 points, which means there were no problems with excessive alcohol consumption. Therefore, the obtained results should be considered inconclusive. Research conducted in Poland and Australia shows that approximately 20–30% of respondents declare higher alcohol consumption compared to the time before the pandemic [6,36,37]. The verification of this hypothesis could also be carried out by examining the described population after the pandemic period and during the return to stabilization of living conditions. Nevertheless, an out-of-pandemic study in Australia found that older people (over 60 years of age) had lower AUDIT scores than other—younger—age groups. The aforementioned study was based on the analysis of cyclically collected data on a representative group of Australians from 2007 to 2016 [38]. Taking into account this study, we can risk a statement that alcohol abuse affects people before the age of 60 more often and that the period of the pandemic does not disturb this long-established relationship. At this point, it should be noted that all studies presented—including the one carried out by the authors—may be subject to some error. Namely, in the case of this project, it should be noted that it was conducted via the Internet and, thus, fewer older people were able to participate, as discussed above. It is worth noting, however, that this limitation appears to affect older people with alcohol abuse problems in particular. Looking at the other studies cited, it can be surmised that carrying out our project via an in-person survey would not change these statistics. This is because it is generally the case that older people (over 60) struggling with alcohol use disorders relatively rarely seek medical help [39].

An important element of this study is also the question about the respondents’ concerns about the COVID-19 pandemic—the answers are summarized in Table 4. The highest percentage of respondents (as many as 73.10%) was most concerned about the health of their relatives. The obtained result fits in the cultural context of Polish society, in which family is still very important to the individual [40]. The next results with the highest number of responses were concerns about the economic crisis (68.71%) and the duration of the pandemic (61.48%), which seems to be fully understood because of the information and forecasts provided by the media and significantly influencing the prediction of the future. It is worth noting that some of these responses may have been due to isolation during the lockdown period.

Our study has certain limitations. As shown in Appendix A, it does not reflect the cross-section of Polish society and cannot be considered representative. This comparison only proves that the interest in participating in the online survey concerns a selected population of people. Most participants in our study are women, which can be explained by many different factors, including the more frequent use of social networks by women through which the survey was conducted. This is evidenced by the report, which was published thanks to the analyses carried out by ComScore in 2010. These data were based on the analysis of the Internet behavior of about 2 million users who had previously consented to the collection of relevant data. This report shows that women spend more time than men browsing social networking sites [41]. It is also worth noting that, in Polish society, there are on average 107 women per 100 men [42]. In our study, this proportion was significantly disturbed, as there are approximately 330 women per 100 men. However, taking into account the purposefulness of our research, the number of people in productive and post-productive age was of greater importance. At this point, it is worth noting that the productive age is in the range of 18–64 years. For this reason, it is reasonable to assume (by analyzing the data from Table 1 and Table 5) that all participants of this study fall within this range. Referring this assumption to the data from the Statistical Yearbook, it is worth noting that only 60.58% of the population is in the productive age [42]. This situation is most likely due to the exclusion criteria for people under the age of 18. In addition, older people (over 64) use social networking sites much less frequently than people of working age. According to the GUS report published in 2019, only 8.9% of people over 65 had basic or secondary digital skills [43]. Conducted research shows that the problem lies not so much in using the Internet as in the impossibility of introducing older people and acquainting them with the regulations of social networks and the principles of their operation [44,45]. According to the results of our study, the place of residence did not affect the distribution of the examined parameters. According to the Central Statistical Office, 60.10% of the population live in cities (of various sizes), and the remaining part—39.90%—lived in villages [18]. Our study was much more frequently attended by the urban population (80.48%). This observation is consistent with the aforementioned GUS report, which indicates that slightly more people in villages and medium-sized towns do not have access to the Internet. According to the report, this is due to “no need to have access to the network at home” [43].

The group of respondents analyzed in this study differs from the structure of the Polish society, which is particularly visible in the aspect of age structure and education. The group of women is clearly over-represented, and the same can be said in the context of the group of people with secondary and higher education. For this reason, this study does not analyze the differences among people with lower than secondary level education. The situation of people living in villages is similar, because they were heavily underrepresented; thus, it is difficult to conduct an analysis for this particular group. This is most probably because the survey was conducted exclusively via the Internet. Such a form resulted from the necessity to keep an appropriate social distance.

## 5. Conclusions

The relationships between the studied parameters do not differ significantly from those presented in the studies outside the COVID-19 pandemic period.People under the age of 50 may be at risk of increased alcohol use in relation to the elderly.Women between the ages of 30 and 49 are most vulnerable to the negative mental health effects of the COVID-19 pandemic.There is a need for further research in which the studied group will obtain a greater degree of compliance with the structure of Polish society.

## Figures and Tables

**Table 1 ijerph-19-07081-t001:** The demographic structure of the studied group.

	Number of Respondents	Percentage Value to the Number of Respondents (%)
Number of respondents	538	100
Sex:		
men	125	23.23
women	413	76.77
Age:		
18–29	366	68.03
30–49	147	27.32
50+	25	4.65
Domicile:		
village	105	19.52
a town with less than 50,000 inhabitants	90	16.73
city with 50,000–200,000 inhabitants	104	19.33
a city with over 200,000 inhabitants	239	44.42
Education:		
primary	7	1.30
vocational	1	0.19
secondary	254	47.21
higher	271	50.37
no answer	5	0.93

**Table 2 ijerph-19-07081-t002:** The results of comparative analyses for the groups of women and men to the tested parameters.

Value	Women *n* = 413		Men *n* = 125		Z	*p*
	Mean	SD	Mean	SD		
Anxiety	7.855	5.187	5.552	5.131	4.779	**<0.001 ***
Consuming alcohol	3.777	3.493	5.640	5.119	−4.417	**<0.001 ***
Generalized aggression	70.395	16.229	69.128	15.506	0.724	0.469
Physical aggression	15.729	5.189	17.776	5.594	−3.910	**<0.001 ***
Verbal aggression	14.521	3.711	14.832	3.587	−0.945	0.344
Anger	18.719	5.888	16.488	5.844	3.474	**<0.001 ***
Hostility	21.426	6.449	20.032	6.244	2.309	**<0.05 ***

SD—standard deviation; *—statistically significant *p* < 0.05.

**Table 3 ijerph-19-07081-t003:** Statistical differences between the study groups divided by sex in terms of anxiety and the level of alcohol consumption after obtaining normalized results.

Value	Women *n* = 413 (100%)	Men *n* = 125 (100%)	*p*
Anxiety			
no anxiety (0–4 points)	114 (27.60%)	66 (52.80%)	**<0.001 ***
mild anxiety (5–9 points)	175 (42.37%)	37 (29.60%)	
moderate anxiety (10–14 points)	75 (18.16%)	14 (11.20%)	
serious anxiety (15–21 points)	49 (11.86%)	8 (06.40%)	
Consuming alcohol			
low risk of dependence (0–7 points)	359 (86.92%)	94 (75.20%)	**<0.01 ***
risky consuming (8–15 points)	50 (12.11%)	24 (19.20%)	
harmful consuming (16–19 points)	2 (00.48%)	4 (03.20%)	
alcohol addiction (20–40 points)	2 (00.48%)	3 (02.40%)	

*—statistically significant *p* < 0.05.

**Table 4 ijerph-19-07081-t004:** Differences between age groups concerning the parameters tested.

Value	18–29 Years Old *n* = 366		30–49 Years *n* = 147		50+ Years *n* = 25		*p*
	Mean	SD	Mean	SD	Mean	SD	
Anxiety	7.393	5.323	7.000	5.071	8.120	5.510	0.617
Consuming alcohol	4.295	4.103	4.313	3.894	2.360	2.596	**<0.01 ***
Generalized aggression	71.691	16.255	67.449	15.027	62.400	15.311	**<0.01 ***
Physical aggression	16.317	5.665	15.918	4.607	16.240	4.772	0.939
Verbal aggression	14.615	3.668	14.755	3.794	13.320	3.051	0.181
Anger	18.568	5.976	17.680	5.952	15.880	4.859	0.052
Hostility	22.191	6.382	19.095	5.642	16.960	7.027	**<0.001 ***

SD—standard deviation; *—statistically significant *p* < 0.05.

**Table 5 ijerph-19-07081-t005:** Statistical differences between the study groups divided by age in terms of anxiety and the level of alcohol consumption after obtaining normalized results.

Value	18–29 Years Old *n* = 366 (100%)	30–49 Years Old *n* = 147 (100%)	50+ Years Old *n* = 25 (100%)	*p*
Anxiety				
no anxiety (0–4 points)	126 (34.43%)	46 (31.29%)	8 (32.00%)	0.598
mild anxiety (5–9 points)	135 (36.89%)	67 (45.58%)	10 (40.00%)	
moderate anxiety (10–14 points)	65 (17.76%)	21 (14.29%)	3 (12.00%)	
serious anxiety (15–21 points)	40 (10.93%)	13 (08.84%)	4 (16.00%)	
Consuming alcohol				
low risk of dependence (0–7 points)	311	119 (80.95%)	23 (92.00%)	0.791
risky consuming (8–15 points)	(84.97%)	25 (17.01%)	2 (08.00%)	
harmful consuming (16–19 points)	47 (12.84%)	2 (01.36%)	0 (00.00%)	
alcohol addiction (20–40 points)	4 (01.09%)	1 (00.68%)	0 (00.00%)	

**Table 6 ijerph-19-07081-t006:** Differences between educational groups concerning the parameters tested.

Value	Medium Level of Education *n* = 254		Higher Level of Education *n* = 271		t	95% CI	*p*
	Mean	SD	Mean	SD			
Anxiety	7.437	5.139	7.266	5.425	0.371	(−0.736)–1.079	0.711
Consuming alcohol	4.315	3.738	4.203	4.283	0.318	(−0.579)–0.803	0.750
Generalized aggression	72.681	15.141	67.823	16.691	3.485	2.120–7.596	**<0.001 ***
Physical aggression	16.535	5.511	15.941	5.254	1.265	(−0.329)–1.517	0.206
Verbal aggression	14.925	3.544	14.332	3.835	1.837	(−0.041)–1.227	0.067
Anger	18.862	5.813	17.686	6.012	2.276	0.161–2.191	**<0.05 ***
Hostility	22.358	6.222	19.863	6.398	4.525	1.412–3.578	**<0.001 ***

SD—standard deviation; CI—confidence interval; *—statistically significant *p* < 0.05.

**Table 7 ijerph-19-07081-t007:** Statistical differences between the study groups divided by level of education in terms of anxiety and the level of alcohol consumption after obtaining normalized results.

Value	Medium Level of Education *n* = 254 (100%)	Higher Level of Education *n* = 271 (100%)	*p*
Anxiety			
no anxiety (0–4 points)	85 (33.46%)	91 (33.58%)	0.805
mild anxiety (5–9 points)	97 (38.19%)	109 (40.22%)	
moderate anxiety (10–14 points)	53 (20.87%)	48 (17.71%)	
serious anxiety (15–21 points)	19 (07.48%)	23 (08.49%)	
Consuming alcohol			
low risk of dependence (0–7 points)	212 (83.46%)	228 (84.13%)	0.813
risky consuming (8–15 points)	38 (14.96%)	36 (13.28%)	
harmful consuming (16–19 points)	2 (00.79%)	4 (01.48%)	
alcohol addiction (20–40 points)	2 (00.79%)	3 (01.11%)	

**Table 8 ijerph-19-07081-t008:** Subjective fears.

Concerns about…	Number of Responses	Percentage of Responses (%)
Total number of responses	1093	100
Family health	799	73.10
Economic crisis	751	68.71
The duration of the pandemic	672	61.48
Financial problems	543	49.68
political security	481	44.01
Difficulties in adapting to the new constraints	477	43.64
own health	423	38.70
The uncertainty of existence	282	25.80
Uncertainty about keeping a job	244	22.32
The meaning of life	219	20.04
deterioration of marital relations	78	7.14
No worries	13	1.19

## Data Availability

The data presented in this study are available upon request from the corresponding author. The data are not publicly available due to private survey research.

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
