# Peer review of "The Influence of Sociodemographic Factors on Symptoms of Anxiety, the Level of Aggression and Alcohol Consumption in the Time of the COVID-19 Pandemic among Polish Respondents"

_ijerph, 2022, doi:10.3390/ijerph19127081_

Round 1

Reviewer 1 Report

Dear authors,

A timely study that measures important issues in context of COVID-19 pandemic. 

As noted by authors, the biggest limitations is lack of random sample, and in comparing distributions to wider Polish society, there are major disparities as a result. This leads to the problem of whether we can generalize these results, and what they might mean. 

One answer is assume they don't mean anything, and simply don't use a recruitment strategy that does not attempt some sort of systematic random sample of a well-defined population.

The other answer is present the results but be very clear and up front about the limitations. I think women are over-represented, as noted, but also middle-upper classes surely over-represented as well due to having enough leisure time and general access to internet. Rural residents also very much under-represented here. So more consideration of how lower class, working class, and rural populations might differ may be important. Further, how might Polish people differ from other world populations, for cultural reasons? What is unique about the culture and politics of Poland, and the specific health measures taken, that might make the reaction here and the behavioural outcomes differ?

Other small point: please ensure that the charts line up as they seemed to be mis-aligned and hence more difficult to read and interpret

An important study, but I do think the recruitment and sampling strategy and treatment of data could have been done carefully so as to avoid the problem of how to generalize the statistical results and what we can ultimately take from this. It is especially hard to make fine distinctions about patterns observed when the sample is not representative of the wider population. 

Author Response

Dear Editor and Reviewers,

We highly appreciate the detailed, valuable comments on our manuscript. The suggestions are very helpful for us, and we have incorporated them into the revised paper.

We have addressed all issues indicated in the review report, and we believe that the revised version meets the journal publication requirements. In the text, all changes are marked with “Track Changes”.

  1. As noted by authors, the biggest limitations is lack of random sample, and in comparing distributions to wider Polish society, there are major disparities as a result. This leads to the problem of whether we can generalize these results, and what they might mean. One answer is assume they don’t mean anything, and simply don’t use a recruitment strategy that does not attempt some sort of systematic random sample of a well-defined population.

Response 1: We are aware that the conclusions of our study do not refer to the broad mass of Polish society only to the population that joined the survey made available online, i.e. predominantly middle-aged and middle-educated women. We signal this shortcoming in the limitations of the study and the conclusion of the paper, drawing attention to the need to broaden the study group to include other social groups in order to obtain a more adequate picture of Polish society. If the epidemic situation allows it, we will continue the study.

  1. The other answer is present the results but be very clear and up front about the limitations. I think women are over-represented, as noted, but also middle-upper classes surely over-represented as well due to having enough leisure time and general access to internet. Rural residents also very much under-represented here. So more consideration of how lower class, working class, and rural populations might differ may be important. Further, how might Polish people differ from other world populations, for cultural reasons? What is unique about the culture and politics of Poland, and the specific health measures taken, that might make the reaction here and the behavioural outcomes differ?

Response 2: Taking into account the reviewer's suggestions, we have significantly expanded the description of limitations of our study by referring to new literature items, which we have added to the bibliography. We have tried to familiarize the Reader with the potential reasons for the over-representation of women with at least secondary education in our study, referring to the specificity of Polish society by providing relevant data. We admit that taking into account our knowledge about the proportion of women in Internet research, we expected a higher share of the female gender, but we did not foresee such a large disproportion. Nevertheless, we believe that the results of our study may prove valuable and motivate us to continue our work.

  1. Other small point: please ensure that the charts line up as they seemed to be mis-aligned and hence more difficult to read and interpret.

Response 3: We have improved the tables to make them more readable.

  1. An important study, but I do think the recruitment and sampling strategy and treatment of data could have been done carefully so as to avoid the problem of how to generalize the statistical results and what we can ultimately take from this. It is especially hard to make fine distinctions about patterns observed when the sample is not representative of the wider population.

Response 4: The research material was collected using convenient sampling via social media. This is a method of data collection, which became particularly popular in the COVID-19 era and is currently used in many scientific studies. Like any method of data collection, it has its limitations - first of all, the difficulty in obtaining a representative sample group, differentiated by strata of society. The adopted attempt to cope with this problem is its awareness and clear presentation of the shortcomings of the method of selection of the study group in the limitations of the work. In addition, the reader has full access to the sociodemographic characteristics of the study group, which makes it easier to assess to which group of people the obtained results refer. 

Once again, we would like to thank you for your insightful and extremely helpful review. We hope that the introduced amendments increased the quality of the manuscript and corresponded to the expectations.

Authors

Reviewer 2 Report

Introduction

The introduction is very short and non-exhaustive. There is a need for a better description of the situation in Poland during the first wave of the pandemic  - as I suppose this is a time of the study. Even IJERPH has published articles that say about Poles' attitudes during the first wave of COVID-19.

The authors should also emphasize that Poles coped with the difficulties caused by the pandemic in various ways. Polish society is religious - does the Covid-19 pandemic have an impact on religiosity in Poland? Does prayer help to deal with psychological stress?

Have there been any problems with access to doctors (psychiatrists, psychologists) in Poland during the first wave of the pandemic?

It is also worth writing something about the attitude of Poles to alcohol? 

Material and methods

Is the survey database publicly available?

Results

Please mind table 3 - consuming alcohol (it is hard to read – something is wrong with the layout).

Discussion

Table 9 should be in the Supplementary Materials, not part of the discussion. What is more, it should be mentioned rather in the Material and Methods or Results, not in the Discussion section.

Line: “It does not reflect the cross-section of Polish society and cannot be considered representative” (185-186) and subsequent lines (187-215) should be in the limitation part at the end of the Discussion section.

Line 274 “The obtained result fits in the cultural context of Polish society, in which the family is still very important to the individual” needs a reference.

Conclusion

Please think about point 2 “People under the age of 50 are the group most exposed to the harmful effects of alcohol” – Is this conclusion supported by your results?

Author Response

Dear Editor and Reviewers,

We highly appreciate the detailed, valuable comments on our manuscript. The suggestions are very helpful for us, and we have incorporated them into the revised paper.

We have addressed all issues indicated in the review report, and we believe that the revised version meets the journal publication requirements. In the text, all changes are marked with “Track Changes”.

  1. The introduction is very short and non-exhaustive. There is a need for a better description of the situation in Poland during the first wave of the pandemic – as I suppose this is a time of the study. Even IJERPH has published articles that say about Poles’ attitudes during the first wave of COVID-19.

Response 1: Thank you for this suggestion. We supplemented the introduction with information on the situation during the first wave of the pandemic in Poland, referring to new literature.

  1. The authors should also emphasize that Poles coped with the difficulties caused by the pandemic in various ways. Polish society is religious – does the Covid-19 pandemic have an impact on religiosity in Poland? Does prayer help to deal with psychological stress?

Response 2: It is difficult for us to answer the question about the influence of religious practices on coping with pandemic stress as well as to answer the question whether the COVID-19 pandemic has somehow affected the level of religiosity in Poland. This is an interesting line of research and we regret that we did not consider this aspect of social life. In the introduction, we have included a handful of information on religiosity.

  1. Have there been any problems with access to doctors ( psychiatrists, psychologists) in Poland during the first wave of the pandemic?

Response 3: Online consultations were widespread and passed the test in our conditions. Until now, this form of contact with the patient has functioned as an alternative to face to face meetings. In the introduction, we included information based on a report by the Polish Patient Ombudsman.

  1. It is also worth writing something about the attitude of Poles to alcohol?

Response 4: Thank you for drawing attention to this aspect. To the best of our ability, we have presented in the introduction data on alcohol consumption by place of residence, gender, age, also referring to the pattern of drinking its consumption in Poland.

  1. Material and methods. Is the survey database publicly available?

Response 5: The link to the survey was available on the Internet for one month. The survey was anonymous, in Polish and addressed to a Polish audience. No one except the first author of the article has access to the compiled results. The data are not made public. They are available on request from the first author of the paper. The acquired data were prepared in accordance with the principles used in scientific research. In the opinion of the Bioethics Committee operating at the Medical University of Silesia in Katowice, consent for the study was not necessary. Moreover, it is worth mentioning that works conducted via the Internet are becoming more and more common and are recognised in the world of science as reliable.

  1. Please mind table 3 – consuming alcohol ( it is hard to read – something is wrong with the layout).

Response 6: We have corrected the table accordingly.

  1. Table 9 should be in the Supplementary Materials, not part of the discussion. What is more, it should be mentioned rather in the Material and Methods or Results, not in the Discussion section. Line: „ It does not reflect the cross-section of Polish society and cannot be considered representative” (185-186) and subsequent lines ( 187-215) should be in the limitation part at the end of the Discussion section.

Response 7: We followed the recommendation. We placed Table 9 in supplementary materials, while its interpretation was lines 185-215, which we placed as suggested in the section discussing limitations of the work.

  1. Line 274 „ The obtained result fits in the cultural context of Polish society, in which the family is still very important to the individual” needs a reference.

Response 8: The relevant item from the available literature has been added to the sentence. Thank you very much for drawing attention to this detail.

  1. Please think about point 2 “ People under the age of 50 are the group most exposed to the harmful effects of alcohol” – is this conclusion supported by your results?

Response 9: The conclusion has been modified accordingly. Point 2 now reads as follows: "People under the age of 50 may be at risk of increased alcohol use in relation to the elderly."

Once again, we would like to thank you for your insightful and extremely helpful review. We hope that the introduced amendments increased the quality of the manuscript and corresponded to the expectations.

Authors

Round 2

Reviewer 1 Report

Thanks for attending to the review comments and improving the paper, well done.

Reviewer 2 Report

The article was well corrected.